# Negative-Pressure Wound Therapy for Prevention of Sternal Wound Infection after Adult Cardiac Surgery: Systematic Review and Meta-Analysis

**DOI:** 10.3390/jcm11154268

**Published:** 2022-07-22

**Authors:** Fausto Biancari, Grazia Santoro, Federica Provenzano, Leonardo Savarese, Francesco Iorio, Salvatore Giordano, Carlo Zebele, Giuseppe Speziale

**Affiliations:** 1Clinica Montevergine, GVM Care & Research, 83013 Mercogliano, Italy; graziasantoro9@gmail.com (G.S.); federfarma@live.it (F.P.); leonardosavarese@libero.it (L.S.); aorticaortic@gmail.com (F.I.); salvatoregiordano970@libero.it (S.G.); carlo.zebele@gmail.com (C.Z.); 2Heart and Lung Center, Helsinki University Hospital, University of Helsinki, 00029 Helsinki, Finland; 3Anthea Hospital, GVM Care & Research, 70124 Bari, Italy; gspeziale@gvmnet.it

**Keywords:** sternal wound infection, surgical site infection, negative-pressure wound therapy, pico, prevena, avelle, vivanotec

## Abstract

The results of current studies are not conclusive on the efficacy of incisional negative-pressure wound therapy (NPWT) for the prevention of sternal wound infection (SWI) after adult cardiac surgery. A systematic review of the literature was performed through PubMed, Scopus and Google to identify studies which investigated the efficacy of NPWT to prevent SWI after adult cardiac surgery. Available data were pooled using RevMan and Meta-analyst with random effect models. Out of 191 studies retrieved from the literature, ten fulfilled the inclusion criteria and were included in this analysis. The quality of these studies was judged fair for three of them and poor for seven studies. Only one study was powered to address the efficacy of NPWT for the prevention of postoperative SWI. Pooled analysis of these studies showed that NPWT was associated with lower risk of any SWI (4.5% vs. 9.0%, RR 0.54, 95% CI 0.34–0.84, I^2^ 48%), superficial SWI (3.8% vs. 4.4%, RR 0.63, 95% CI 0.29–1.36, I^2^ 65%), and deep SWI (1.8% vs. 4.7%, RR 0.46, 95% CI 0.26–0.74, I^2^ 0%), but such a difference was not statistically significant for superficial SWI. When only randomized and alternating allocated studies were included, NPWT was associated with a significantly lower risk of any SWI (3.3% vs. 16.5%, RR 0.22, 95% CI 0.08–0.62, I^2^ 0%), superficial SWI (2.6% vs. 12.4%, RR 0.21, 95% CI 0.06–0.69, I^2^ 0%), and deep SWI (1.2% vs. 4.8%, RR 0.17, 95% CI 0.03–0.95, I^2^ 0%). This pooled analysis showed that NPWT may prevent postoperative SWI after adult cardiac surgery. NPWT is expected to be particularly useful in patients at risk for surgical site infection and may significantly reduce the burden of resources needed to treat such a complication. However, the methodology of the available studies was judged as poor for most of them. Further studies are needed to obtain conclusive results on the potential benefits of this preventative strategy.

## 1. Introduction

Sternal wound infection (SWI) after adult cardiac surgery is associated with an increased risk of early and late mortality, prolonged hospital stay, and a significant resource burden for its treatment [1,2]. Coronary artery bypass grafting is a procedure at high risk for such a complication, particularly when both internal mammary grafts are used [3,4]. Still, patients undergoing valve surgery are not spared by SWI. The skeletonization of the internal mammary artery [5], use of topical antibiotics [6], and tight perioperative glycemic control [6] are some of the measures adopted to reduce the risk of SWI, but this complication may still develop in a significant number of patients, particularly in those with multiple comorbidities [7]. Recent meta-analyses reported on a certain benefit of NPWT for the prevention of surgical site infection following different types of surgical procedures [8,9]. However, pooled data on the impact of NPWT for the prevention of surgical site infection after adult cardiac surgery is lacking; despite this, SWI is the most dreaded postoperative infective complication. In this study, we aimed to pool the data from current randomized and observational studies on the impact of NPWT in preventing SWI in patients undergoing adult cardiac surgery.

## 2. Material and Methods

The present systematic review and meta-analysis is registered in the international prospective register of systematic reviews PROSPERO with the reference code CRD42022320120.

### 2.1. Literature Search Strategy

PubMed, Scopus, and Google were searched on March 2022 for any studies which evaluated the efficacy of NPWT in preventing SWI in patients who underwent adult cardiac surgery. The terms used for this literature search were the following: ‘‘negative pressure” and “sternal”, “sternum”, “sternotomy”, “surgical site infection”, and “wound infection” OR “coronary”. The abstracts of retrieved studies were evaluated and, if fulfilling the inclusion criteria, the full text of each study was independently evaluated by two coauthors (Biancari F., Santoro G.) for inclusion in this meta-analysis. Reference lists of articles on this topic were searched as well. We applied the guidelines for Preferred Reporting Items for Systematic reviews and Meta-Analyses (PRISMA) [10] (Appendix A).

### 2.2. Treatment Definition and Inclusion/Exclusion Criteria

Eligible studies were those reporting on the outcomes of patients who underwent any adult cardiac surgery procedure. The main inclusion criteria are summarized according to the Population, Interventions, Comparison and Outcomes (PICO) criteria (Table 1).

Studies fulfilling the following inclusion criteria were entered in this analysis: (1) provide data on patients who underwent adult cardiac surgery; (2) provide data on negative-pressure wound therapy for the prevention of sternal wound infection; (3) provide data on standard sternal wound dressing; (4) provide data on postoperative surgical site infection of the sternal wound and the mediastinum; (5) include patients aged 18 years or older; (6) be a prospective or retrospective observational study; (7) be published in English language as a full article; and (8) be published later than 2000.

Articles were not eligible for study inclusion in the case of any of the following criteria: (1) data are not clear or inaccurate; (2) no information on surgical site infection; (3) data presented only in the abstract; (4) lack of comparative data on standard wound therapy; and (5) article published in a non-English language.

### 2.3. Data Extraction

Data were independently collected by two investigators (Santoro G. and Biancari F.) and checked by two investigators (F.P. and F.I.). Disagreement on collected data was settled by consensus between these investigators. Missing data or other information were not asked from the authors except for studies from colleagues working within our research network. Characteristics of the included studies and patients’ clinical variables were retrieved from each article (Appendix A). Two investigators (Santoro G. and Biancari F.) assessed the quality of the studies using the National Heart, Blood, and Lung Institute (NHBLI) criteria for study quality assessment of case–control series (https://www.nhlbi.nih.gov/health-topics/study-quality-assessment-tools; accessed on 23 March 2022) (Appendix A).

### 2.4. Outcome Measures

The primary outcome of this analysis was any SWI. Secondary outcomes were superficial SWI and deep SWI. The definition criteria of these outcomes were those originally reported in each study.

### 2.5. Statistical Analysis

Considering an anticipated heterogeneity among observational studies, absolute values and means were pooled using random-effects models. Heterogeneity of outcomes across studies was evaluated using the I^2^ test, with I^2^ < 40% indicating non-significant heterogeneity. Publication bias was evaluated assessing funnel plots. The results are expressed as pooled untransformed proportions and means. Risk of SWI was summarized as risk ratio (RR) and 95% confidence interval (95% CI). Leave-one-out meta-analysis was performed to confirm the consistency of the overall analysis. This method implies performing multiple meta-analyses by excluding one study at each analysis and it is useful to assess the impact of each study on the overall risk estimate. Data are summarized in tables, forest plots, and funnel plots. Sensitivity analysis was performed according to quality of the included studies. Statistical analysis was performed using the RevMan (Review Manager Web, Version 1.22.0. The Cochrane Collaboration, 2020, available at: https://training.cochrane.org/online-learning/core-software/revman/revman-5-download, accessed on 23 March 2022) and Open Meta-Analyst (Brown University, Providence, RI, USA; available at: http://www.cebm.brown.edu/openmeta/, accessed on 23 March 2022) software. A *p* < 0.05 was considered statistically significant.

## 3. Results

One hundred and ninety-one studies were retrieved from a systematic literature search (Appendix A). Ten studies [11,12,13,14,15,16,17,18,19,20] fulfilled the inclusion criteria. Their data are summarized in Table 2 and Appendix A. Two studies were randomized trials, one study adopted alternating allocation and seven studies were non-randomized. Two studies used propensity-score matching to adjust for selection bias. The quality of these studies was fair for three studies and poor for seven studies (Appendix A). Study heterogeneity was low for the pooled analysis of deep SWI (I^2^ 0%), but not for studies evaluating superficial SWI (I^2^ 65%) and any SWI (I^2^ 48%) (Figure 1). Similarly, the funnel plot was symmetrical for studies evaluating deep SWI, and asymmetrical for studies evaluating any SWI and superficial SWI (Appendix A).

NPWT was associated with a lower risk of any SWI (nine studies: pooled rates 4.5% vs. 9.0%, RR 0.54, 95% CI 0.34–0.84, I^2^ 48%), superficial SWI (eight studies: pooled rates 3.8% vs. 4.4%, RR 0.63, 95% CI 0.29–1.36, I^2^ 65%), and deep SWI (nine studies: pooled rates 1.8% vs. 4.7%, RR 0.46, 95% CI 0.26–0.74, I^2^ 0%) (Figure 1), but such a difference was not statistically significant for superficial SWI. These findings were confirmed in leave-one-out meta-analysis, which showed the consistency of the overall risk estimate by excluding one study at a time for any SWI and deep SWI, but not for superficial SWI (Appendix A).

When only randomized and alternating allocated studies were included in the analysis, NPWT was associated with a significantly lower risk of any SWI (two studies [14,18], 115 patients in the NPWT group and 115 patients in the control group; pooled rates 3.3% vs. 16.5%, RR 0.22, 95% CI 0.08–0.62, I^2^ 0%), superficial SWI (two studies [14,18], 115 patients in the NPWT group and 115 patients in the control group; pooled rates 2.6% vs. 12.4%, RR 0.21, 95% CI 0.06–0.69, I^2^ 0%), and deep SWI (three studies [12,14,18], 161 patients in the NPWT group and 167 patients in the control group; pooled rates 1.2% vs. 4.8%, RR 0.17, 95% CI 0.03–0.95, I^2^ 0%).

## 4. Discussion

The results of the present pooled analysis could be summarized as follows: (1) there is evidence from recent studies of a widespread use of NPWT in adult cardiac surgery [11,13,16]; (2) the efficacy of NPWT in preventing post-sternotomy SWI has not been widely investigated; (3) NPWT may decrease the risk of SWI after adult cardiac surgery and its benefit may be more pronounced in preventing deep SWI; and (4) still, the evidence of a beneficial effect of NPWT in this patient population is based on studies with suboptimal quality.

The findings of randomized or alternating allocation studies seems to confirm the results of observational studies which, with a low grade of heterogeneity, indicate that NPWT is a valid preventive measure to avoid post-sternotomy SWI. We speculate that the use of NPWT may be most effective for the prevention of deep SWI with a net benefit for the overall rates of any SWIs. This is confirmed by the results of most studies, with few exceptions [11]. A few recent studies [11,13,16] showed that there is a widespread use of this treatment, which suggest that clinicians are confident of a net benefit of NPWT, particularly in patients at highest risk of surgical site infection [11,13]. Since the risk of SWI is particularly increased after bilateral internal mammary artery grafting [21], a benefit with the preventative use of NPWT may be more evident in such high-risk patients. This may be particularly relevant when bilateral internal mammary artery grafting is used in women and in patients with obesity, diabetes, chronic renal failure, pulmonary disease, atrial fibrillation, and/or critical preoperative conditions [8,21]. The limited number of studies and incomplete data included in this meta-analysis do not allow sensitivity analyses on these relevant comorbidities.

It is worth noting that the results of the study by Myllykangas et al. [11] differ markedly from the overall risk of SWI herein estimated. The authors should be congratulated for having included in their analysis a large series of patients with rather well-balanced study cohorts after propensity score matching. However, the NPWT cohort was prospectively evaluated from 2018 to 2020, while data on the standard wound dressing cohort were retrospectively collected from patients operated on from 2012 to 2017. We may expect that the prospective nature of the NPWT cohort might have led to an increased detection rate of SWI, particularly those not requiring surgical debridement, compared to the historical control cohort.

According to the pooled rates herein calculated from all included studies, we estimated that 22.2 patients should be treated with NPWT to avoid a single event of any SWI, whilst the number of patients needed to be treated is 34.5 to avoid a single event of deep SWI. Although the number of patients to be treated can be considered quite large, the potential benefit of NPWT in preventing SWI should be viewed in the context of the increased burden of resources for the prolonged treatment of this complication [22], as well as the significant clinical implication of post-sternotomy SWI, which translates into an increased risk of mortality [1].

Considering the costs of these devices, NPWT may not be cost-effective when used routinely [23]. Hawkins et al. [23] estimated that this treatment method may be cost-effective only when the expected deep SWI rate is higher than 1.3%. This means that less than half of patients undergoing cardiac surgery with comorbidities associated with SWI [8,24] may clinically benefit from NPWI with an expected saving of resources. Future studies should investigate the efficacy of NPWT according to the individual patient’s risk profile. This issue could not be addressed in this meta-analysis, although several of the included studies have investigated patients at a high risk for surgical site infection (Appendix A).

### Study Limitations

The suboptimal quality of the available studies is the main limitation of the present pooled analysis. However, the heterogeneity of the included studies is rather limited, and this finding suggests a potential benefit of NPWT, particularly in preventing deep SWI. Furthermore, a balance in the preoperative risk of these patients can be observed in terms of the pooled prevalence of baseline comorbidities of the study cohorts (Table 2). Second, beside the lack of data in terms of adequately adjusted analyses for potential confounders, these findings suggest that only one of the current studies might have been powered to address the efficacy of this treatment for the prevention of SWI. Indeed, to confirm a reduction in the incidence of SWI from 9.0% to 4.5%, as observed in this analysis, we estimated that 487 patients should be included in each study group, while 586 patients in each group should be investigated to confirm a reduction in deep SWI rate from 4.7% to 1.8% (alpha 0.050; power 0.80). Third, the definition criteria of SWI differed between studies and there is no information regarding the assessors of this complication. Fourth, the limited number of studies and incomplete data of studies included in this meta-analysis do not allow sensitivity analyses of comorbidities, which are known to increase the risk of SWI. Finally, inter-institutional differences in terms of clinical pathways, patients’ characteristics, the perioperative management of blood glucose levels, and methods adopted intraoperatively for the prevention of SWI may further introduce bias in the present analysis. Such issues should be considered when planning future studies on incisional NPWT.

## 5. Conclusions

This systematic review showed that there are a limited number of studies on the potential benefits of incisional NPWT for the prevention of SWI after adult cardiac surgery. Furthermore, the quality of the available studies is suboptimal to obtain conclusive results on this issue. Still, the present analysis showed that, based on current data, NPWT may prevent postoperative SWI after adult cardiac surgery. NPWT is expected to be particularly useful in patients at risk of surgical site infection and may significantly reduce the burden of resources needed to treat such a complication. In view of the potential impact of this method to prevent such a severe postoperative complication, the efficacy of NPWT to prevent SWIs should be assessed in large clinical trials.

## Figures and Tables

**Figure 1 jcm-11-04268-f001:**
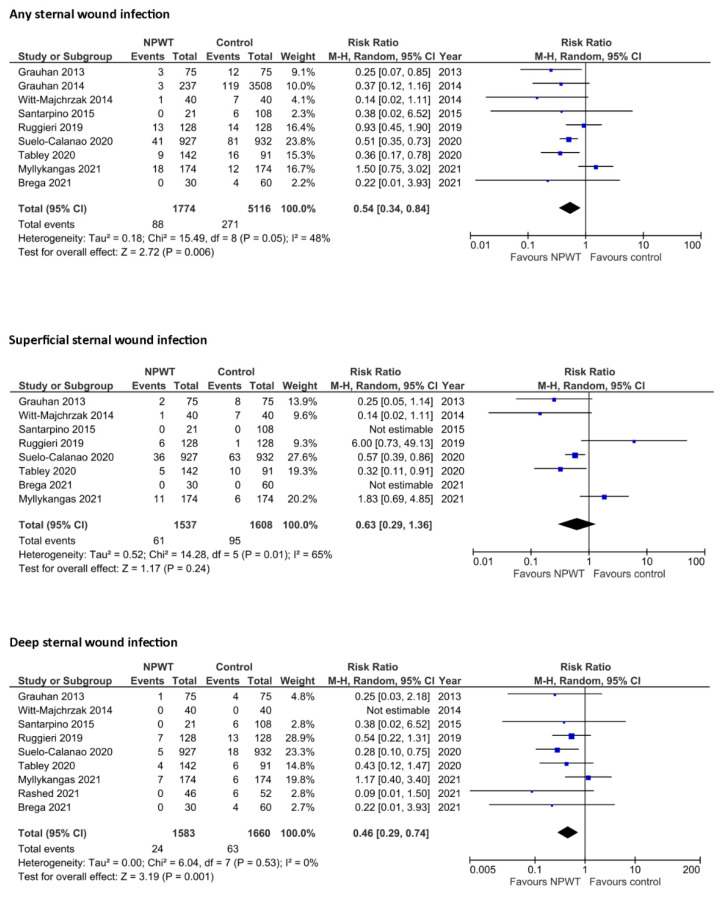
Forest plots of study interventions and outcomes [11,12,13,14,15,16,17,18,19,20].

**Table 1 jcm-11-04268-t001:** Population, intervention, comparison and outcomes (PICO) of the present meta-analysis.

PICO	Description
Population	Patients who underwent any adult cardiac surgery procedure
Intervention	Negative-pressure wound therapy with any commercially available device
Comparison	Standard wound therapy
Outcomes	Any sternal wound infection, superficial sternal wound infection, deep sternal wound infection

**Table 2 jcm-11-04268-t002:** Pooled characteristics of patients of the included studies.

Variables	No. of Studies	NPWT Cohort	Control Cohort	*p*-Value
Age, mean in years	9	66.1	66.2	0.975
Female	8	29.5%	28.4%	0.937
BMI > 30 kg/m^2^	6	59.4%	64.8%	0.785
Diabetes	9	49.4%	52.3%	0.825
Pulmonary disease	9	15.9%	17.3%	0.992
Any coronary surgery	9	85.7%	84.1%	0.354
Isolated coronary surgery	8	75.8%	76.1%	0.862

Means and proportions are pooled estimates. NPWT—negative-pressure wound therapy. *p*-values are from odds ratios estimates.

## Data Availability

Not applicable.

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
