# Peer review of "Negative-Pressure Wound Therapy for Prevention of Sternal Wound Infection after Adult Cardiac Surgery: Systematic Review and Meta-Analysis"

_jcm, 2022, doi:10.3390/jcm11154268_

Round 1
Reviewer 1 Report
To the authors
In the manuscript jcm-1813585 entitled Negative pressure wound therapy for prevention of sternal wound infection after adult cardiac surgery: systematic review and meta-analysis, Dr. Biancari and colleagues investigated a meta-analysis from 10 published studies related to negative pressure wound therapy (NPWT) on adult cardiac surgery. Target outcomes were defined as any sternal wound infection (SWI), superficial SWI and deep SWI. In the main results, pooled analysis revealed that NPWT had significant preventive effect in any SWI and deep SWI, but did not in superficial SWI.
It seems understandable that SWI is sometimes useful in cases after cardiac surgery, but the authors should describe the novelties of the current study more precisely including what the differences were from previous meta-analyzed reports.
Major critiques are described below.
1. NPWT or vacuum-assisted closure (VAC) therapy for a prevention of sternal wound infection had been well investigated including meta-analysis assessment. Raja et al. reported a review from 13 papers which showed VAC therapy provides a viable and efficacious adjunctive method to treat postoperative wound infection after cardiac surgery (Interact Cardiovasc Thorac Surg. 2007, PMID: 17669926). Falagas et al. also reported a meta-analysis from 22 papers which showed VAC therapy was associated with lower mortality than other surgical techniques for patients with sternal wound infections after cardiothoracic surgery (PLoS One. 2013, PMID: 23741379). Both systematic reports concluded that VAC therapy had limited advantages for the management of wound infection, but also claimed that routine use of VAC therapy was not recommended and that further prospective randomized studies were needed. What is novel and different in the current investigation compared with the findings and conclusions in the previous studies?
2. Whereas some characteristics of the selected studies were tabulated in the Supplementary table 2, it seems short for enough description. The authors should consider to add some more demographic and clinical characteristics of the study populations such as patient number, age, sex, BMI, used surgical technique (including the use of bilateral internal mammary artery [BIMA]) and key comorbidities (including DM, AF and CKD).
3. The discussion described in the manuscript is fair but limited to general findings. Although the authors noted only “there is a widespread use of this treatment....particularly in patients at highest risk of surgical site infection (Line 142-144)”, it should be important to discuss who particularly should be considered to be apply NPWT on cardiac surgery. Davierwala et al. reviewed about BIMA grafting for CABG which seems one of the key factors for postoperative deep SWI and described that some additional factors should associate with the occurrence of SWI (Int J Surg. 2015, PMID: 25612853). These aspects should be also discussed in the current investigation.
4. The authors noted “Study heterogeneity was low for pooled analysis of any SWI and deep SWI, but not for studies evaluating superficial SWI (Fig. 1, Suppl. Fig. 2-4). (Line 111-113)”, but the issues assessed by the results in Fig 1 and Fig S2-S4 should be different. The term “study heterogeneity” seems obscure, and the authors should consider rephrasing it into “heterogeneity of study outcomes” or something. The aims of presenting Fig S2-S4 should be the evaluation for publication bias. State the aims and the results of the assessments precisely and clearly.
5. Sensitivity analysis was performed using leave-one-out analyses and shown in Figure S5. Because of the lack of legends, the method and assessment details for leave-one-out cross-validation (LOOCV) were unclear. It seems that the robustness of results was shown in any SWI and deep SWI but not in superficial SWI. The authors should describe these assessments more clearly.
6. Whereas the results of sensitivity analysis were fair, it should be noted that one study named “Myllykangas 2021” (might be Ref#22, Thorac Cardiovasc Surg. 2022, PMID: 34521138), different from all the other studies, showed unfavorable result for any SWI in Figure 1. It might be valuable to discuss why the result from this study had been oppose to other studies, which could deeply understand the advantages and disadvantages of NPWT for cardiac surgery.
Minor comment
Line 116-121
NPWT was associated with lower risk of any SWI (9 studies: pooled rates 4.5% vs. 9.0%, RR 0.54, 95%CI 0.34-0.84, I2 48%) (Fig. 1), superficial SWI (8 studies: pooled rates 3.8% vs. 4.4%, RR 0.63, 95%CI 0.29-1.36, I2 65%) (Fig. 2) and deep SWI (9 studies: pooled rates 1.8% vs. 4.7%, RR 0.46, 95%CI 0.26-0.74, I2 0%) (Fig. 3), but such a difference was not statistically significant for superficial SWI. These findings were confirmed in leave-one-out meta-analysis (Suppl. Fig. 5).
There are no Figures named Figure 2 and Figure 3 in the current manuscript.
Author Response
1. NPWT or vacuum-assisted closure (VAC) therapy for a prevention of sternal wound infection had been well investigated including meta-analysis assessment. Raja et al. reported a review from 13 papers which showed VAC therapy provides a viable and efficacious adjunctive method to treat postoperative wound infection after cardiac surgery (Interact Cardiovasc Thorac Surg. 2007, PMID: 17669926). Falagas et al. also reported a meta-analysis from 22 papers which showed VAC therapy was associated with lower mortality than other surgical techniques for patients with sternal wound infections after cardiothoracic surgery (PLoS One. 2013, PMID: 23741379). Both systematic reports concluded that VAC therapy had limited advantages for the management of wound infection, but also claimed that routine use of VAC therapy was not recommended and that further prospective randomized studies were needed. What is novel and different in the current investigation compared with the findings and conclusions in the previous studies?
Response: We thank the Reviewer for her/his comments. The main difference between the cited studies on VAC therapy and the present pooled analysis resides either in the indication for these treatment methods and the treatments themselves. In the studies by Falagas et al. and Raja et al., VAC therapy was used as a treatment for sternal wound infection. In our study, NPWT was used to prevent sternal wound infection. Even if both treatment methods are based on negative pressure suction, the devices are different: VAC device is applied with a sponge on an infected, open surgical wound, while NPWT is a wound dressing applied on clean, closed wound.
Changes: None.
2. Whereas some characteristics of the selected studies were tabulated in the Supplementary table 2, it seems short for enough description. The authors should consider to add some more demographic and clinical characteristics of the study populations such as patient number, age, sex, BMI, used surgical technique (including the use of bilateral internal mammary artery [BIMA]) and key comorbidities (including DM, AF and CKD).
Response: We do agree with the Reviewer on the need to present patients´data. Please, note the data on renal failure could not be extracted due to differences in the definition criteria of this comorbidity.
Changes: We added a Supplementary table 3 which summarizes patients´characteristics included in the available studies.
3. The discussion described in the manuscript is fair but limited to general findings. Although the authors noted only “there is a widespread use of this treatment....particularly in patients at highest risk of surgical site infection (Line 142-144)”, it should be important to discuss who particularly should be considered to be apply NPWT on cardiac surgery. Davierwala et al. reviewed about BIMA grafting for CABG which seems one of the key factors for postoperative deep SWI and described that some additional factors should associate with the occurrence of SWI (Int J Surg. 2015, PMID: 25612853). These aspects should be also discussed in the current investigation.
Response: We do agree with the Reviewer on the importance of risk factors which indicate the use of preventative strategies against postoperative sternal wound infection and we added two paragraphs discussing these issues.
Changes: We added these comments to the Discussion: “Since the risk of SWI is particularly increased after bilateral internal mammary artery grafting (21), a benefit with the preventative use of NPWT may be more evident in such high-risk patients. This may be particularly relevant when bilateral internal mammary artery grafting is used in women and in patients with obesity, diabetes, chronic renal failure, pulmonary disease, atrial fibrillation and/or critical preoperative conditions (8,21).”. We added another sentence to the Limitations section: “The limited number of studies and incomplete data of studies included in this meta-analysis does not allow sensitivity analyses of comorbidities which are known to increase the risk of SWI.”.
4. The authors noted “Study heterogeneity was low for pooled analysis of any SWI and deep SWI, but not for studies evaluating superficial SWI (Fig. 1, Suppl. Fig. 2-4). (Line 111-113)”, but the issues assessed by the results in Fig 1 and Fig S2-S4 should be different. The term “study heterogeneity” seems obscure, and the authors should consider rephrasing it into “heterogeneity of study outcomes” or something. The aims of presenting Fig S2-S4 should be the evaluation for publication bias. State the aims and the results of the assessments precisely and clearly.
Response: We appreciate the Reviewer´s comments on these mistakes.
Changes: We added to the Methods and Results sections a description of the methodology regarding heterogeneity and funnels plots and we rephrased a sentence of the Results section. In the Methods section: “Heterogeneity of outcomes across studies was evaluated using the I2 test, with I2 <40% indicating non-significant heterogeneity. Publication bias was evaluated assessing funnel plots.”. In the Results section: “Study heterogeneity was low for pooled analysis of deep SWI (I2 0%), but not for studies evaluating superficial SWI (I2 65%) and any SWI (I2 48%) (Fig. 1). Similarly, funnel plot was symmetrical for studies evaluating deep SWI, and asymmetrical for studies evaluating any SWI and superficial SWI (Suppl. Fig. 2-4).”.
5. Sensitivity analysis was performed using leave-one-out analyses and shown in Figure S5. Because of the lack of legends, the method and assessment details for leave-one-out cross-validation (LOOCV) were unclear. It seems that the robustness of results was shown in any SWI and deep SWI but not in superficial SWI. The authors should describe these assessments more clearly.
Response: We thank the Reviewer for commenting on the leave-one-out analysis. We believe that the information suggested by the Reviewer is relevant for a throughout understanding of these results.
Changes: We added the following sentences to the Methods and results sections, respectively: “This method implies performing multiple meta-analyses by excluding one study at each analysis and it is useful to assess the impact of each study on the overall risk estimate”, “which showed consistency of the overall risk estimate by excluding one study at time for any SWI and deep SWI, but not for superficial SWI”.
6. Whereas the results of sensitivity analysis were fair, it should be noted that one study named “Myllykangas 2021” (might be Ref#22, Thorac Cardiovasc Surg. 2022, PMID: 34521138), different from all the other studies, showed unfavorable result for any SWI in Figure 1. It might be valuable to discuss why the result from this study had been oppose to other studies, which could deeply understand the advantages and disadvantages of NPWT for cardiac surgery.
Response: We do agree on the need to discuss the findings of the Myllykangas´study in the light of these pooled results.
Changes: We added the following paragraph to the Discussion section.
7-8. Line 116-121. NPWT was associated with lower risk of any SWI (9 studies: pooled rates 4.5% vs. 9.0%, RR 0.54, 95%CI 0.34-0.84, I2 48%) (Fig. 1), superficial SWI (8 studies: pooled rates 3.8% vs. 4.4%, RR 0.63, 95%CI 0.29-1.36, I2 65%) (Fig. 2) and deep SWI (9 studies: pooled rates 1.8% vs. 4.7%, RR 0.46, 95%CI 0.26-0.74, I2 0%) (Fig. 3), but such a difference was not statistically significant for superficial SWI. These findings were confirmed in leave-one-out meta-analysis (Suppl. Fig. 5).
There are no Figures named Figure 2 and Figure 3 in the current manuscript.
Response: We thank this Reviewer for this remark.
Changes: We have eliminated Fig. 2 and Fig. 3 as they are merged in a single one (i.e. Figure 1).
Reviewer 2 Report
The manuscript by Biancari et al. is a systematic review about the prevention of sternal wound infection post cardiac surgery in adults. The article is well written and up-to-date and can be published after minor modifications as suggested below.
1) The abstract needs to be improved. Please mention the results of the current studies and why are they inconclusive.
2) I suggest also to expand the conclusion section, with a more comprehensive explanation of the possible applications of the discussed data in clinical practice in the future.
Author Response
1. The abstract needs to be improved. Please mention the results of the current studies and why are they inconclusive.
Response: We do agree with the Reviewer on the need to improve the Abstract section. We stated that the methodology of the included studies was poor in seven of them. We added a comment also on the limited sample size. Please note the limited number of words allowed for the Abstract by the Journal.
Changes: We added the following sentence to the Result section of the Abstract: “Only one study was powered to address the efficacy of NPWT for prevention of postoperative SWI.”.
2. I suggest also to expand the conclusion section, with a more comprehensive explanation of the possible applications of the discussed data in clinical practice in the future.
Response: We do agree with the Reviewer on the need to improve the Conclusions section.
Changes: We added the following sentence to the Conclusions section (both in the Abstract and text): “NPWT is expected to be particularly useful in patients at risk for surgical site infection and may significantly reduce the burden of resources needed to treat such a complication”.
Reviewer 3 Report
The topic is important and the idea to create such a paper is very good, I believe. We have some limited experience with the SWI prevention by NPWT, and it works.
Prof. Panayotov
Round 2
Reviewer 1 Report
To the authors
The reviewer thanks the authors’ effort to thoroughly revise the manuscript along with the reviewer’s comments. All the concerns were solved to be enough satisfied, and the reviewer believes that the manuscript comes up to be more qualified.